# Structure of the Polysaccharide Secreted by *Vibrio alginolyticus* CNCM I-5035 (Epidermist 4.0^TM^)

**DOI:** 10.3390/md18100509

**Published:** 2020-10-09

**Authors:** Sophie Drouillard, Rémi Chambon, Isabelle Jeacomine, Laurine Buon, Claire Boisset, Anthony Courtois, Bertrand Thollas, Pierre-Yves Morvan, Romuald Vallée, William Helbert

**Affiliations:** 1Centre de Recherches sur les Macromolécules Végétales (CERMAV), Université Grenoble Alpes, CNRS, 38000 Grenoble, France; sophie.drouillard@cermav.cnrs.fr (S.D.); remichambon@yahoo.fr (R.C.); isabelle.jeacomine@cermav.cnrs.fr (I.J.); laurine.buon.drouillard@cermav.cnrs.fr (L.B.); claire.boisset-helbert@cermav.cnrs.fr (C.B.); 2Polymaris Biotechnology, 160 Rue Pierre Rivoalon, 29200 Brest, France; Anthony.courtois@polymaris.com (A.C.); Bertrand.thollas@polymaris.com (B.T.); 3CODIF International, 35400 Roz-sur-Couesnon, France; py.morvan@codif.com (P.-Y.M.); r.vallee@codif.com (R.V.)

**Keywords:** exopolysaccharide, structure, NMR, *Vibrio alginolyticus*, Epidermist

## Abstract

*Vibrio alginolyticus* (CNCM I-5035) secretes an exopolysaccharide used as ingredient in cosmetic industry under the trademark Epidermist 4.0^TM^. It is appreciated for its ability to improve the physical and chemical barrier functions of the skin by notably increasing the keratinocyte differentiation and epidermal renewal. Composition analyses and in depth characterization of the polysaccharides as well as oligosaccharides obtained by mild acid hydrolyses revealed that it was composed of a repetition unit of three residues: d-galactose (d-Gal), d-*N*-acetylglucosamine (GlcNAc) and l-*N*-acetylguluronic acid, of which 30% (M/M) was acetylated in position 3. The complete structure of the polysaccharide was resolved giving the repetition unit: [→3)-α-d-Gal-(1→4)-α-l-GulNAcA/α-l-3OAc-GulNAcA-(1→4)-β-d-GlcNAc-(1→].

## 1. Introduction

Polysaccharide are very complex and diverse macromolecules attested by the very high number of structures―19.773 entries to date―listed in the “Carbohydrate Structure Databank” (http://csdb.glycoscience.ru/database/index.html) [1,2,3]. The very wide structural diversity is explained by the stereochemistry of carbohydrate and the numerous possibilities of linkages between residues [4]. In addition to the complexity of the carbohydrate backbone, polysaccharides are often decorated by organic (e.g. lactate, acetate, amino acids) and inorganic (e.g. sulfate, phosphate) derivatives, therefore increasing the number of possible structures.

Many microorganisms, including marine bacteria, secrete extracellular polysaccharides, called exopolysaccharides (EPSs) [5,6]. Despite the numerous polysaccharide structures elucidated, the diversity of EPS seems largely underestimated and tools to predict their structures are still absent. The massive sequencing of DNA of isolated organisms (genomic) or extracted from the environment (metagenomics), did not allow the emergence of bioinformatic methods helping in the prediction of the structures of the biosynthesized polysaccharides. Therefore, solving the structure of the polysaccharides by chemical and spectroscopic methods follows a unique strategy. Because these approaches are fastidious, structural analyses are usually targeted on polysaccharides that present identified interest in the understanding of their biological properties in vivo (e.g., antigenic properties), ex vivo (e.g., cosmeceutical properties), and their gelling or thickening properties.

In contrast to that of plant and algal polysaccharides, large-scale production of marine EPSs can be easily controlled independently of seasonal variation, for example. In addition, because of their very high molecular weight, EPSs can be easily separated from other molecules and be obtained at very high grade of purity. These technical advantages are not sufficient for the development of novel marine EPSs as gelling or thickening agents which are facing the already marketed terrestrial bacterial EPSs, such as xanthan (*Xanthomonas campestris*) or gellan (*Sphingomonas paucimobilis*) Recently, marine EPSs showing both interesting rheological properties and biological properties could be valorized in niche applications in the biomedical and cosmetic sectors. Among them, the elucidation of the structures of two EPSs secreted by two distinct *Vibrio alginolyticus* strains used as bioactive ingredients in cosmetic applications revealed original structures presenting decoration of the polysaccharide backbone by amino acids (strain CNCM I-4994) and the occurrence of the rare “nosturonic acid” residue (strain CNCM I-5034) [7,8].

In this line of investigations, the polysaccharide secreted by *Vibrio alginolyticus* CNCM I-5035 was recently developed as an ingredient in the cosmetics industry under the trademark Epidermist 4.0^TM^ [9]. The polysaccharide has numerous beneficial effects on human skin validated in vitro and in vivo. Notably, it improves the physical and chemical barrier functions, by increasing the keratinocyte differentiation and epidermal renewal, and by increasing the immune defence against the infectious agents and pathogens involved in acne like *Cutibacterium acnes* (formerly *Propionibacterium acnes*). As for the previous *V. alginolyticus*, we have undertaken the complete structural analysis of the EPS, which revealed the occurrence of the rare 3O-acetyl-2-acetamido-2-deoxy-〈-l-guluronic acid and 2-acetamido-2-deoxy-〈-l-guluronic acid residues.

## 2. Results and Discussion

### 2.1. Composition of the Polysaccharide

Size-exclusion chromatography (SEC) coupled to multi-angle laser light scattering showed that the purified EPS was composed of one species of molecule having a molecular weight (MW) of 5.5 × 10^5^ Da with a narrow polydispersity index of 1.025. Composition analysis using gas chromatography (GC), after complete hydrolysis of the polysaccharide and derivatization of the products, revealed galactose (Gal) and glucosamine (GlcN). Some signals of the chromatogram could not be attributed at this stage using standard residues available in the laboratory suggesting the occurrence of, at least, one additional residue (Appendix A). Elemental analysis did not uncover any inorganic derivatives such as sulfate or phosphate ester groups. Absolute configuration analyses demonstrated that the two residues identified belonged to the d-series (Appendix A).

The ^1^H NMR spectrum of the polysaccharide (Figure 1A) showed four anomeric signals that we attributed to A, B, B’ and C residues (A-H1 = 5.13 ppm; B-H1 = 5.14, B’-H1:5.18 and C-H1 = 4.81 ppm). We noticed also three methyl groups attributed to three different acetyl groups (between 2.0 ppm and 2.3 ppm). The ring protons of the 〈-linked residue A were assigned, starting from the anomeric proton A-H1, by successfully combining correlation spectroscopy (COSY, Appendix A) and total COSY (TOCSY) experiments. The chemical shifts of the corresponding carbons were determined using heteronuclear single-quantum correlation experiments (HSQC, Appendix A) and are reported in Table 1. The chemical shifts recorded were in agreement with the 〈-d-galactose. The residue C corresponded to the ^®^-linked *N*-acetyl-glucosamine (C-H1 = 4.81 ppm; C-C1 = 103.19 ppm). The proton and carbon chemical shifts were also determined, highlighting the characteristic C-H2 = 3.81 ppm and C-C2 = 56.84 ppm chemical shift of the amine residue (Table 1).

After an alkaline treatment of the polysaccharide, the ^1^H spectrum (Figure 1B) was simplified and three anomeric signals instead of four were then observed (A-H1 = 5.14 ppm; B-H1 = 5.14 ppm and C-H1 = 4.81 ppm). This alkaline treatment leads to the removal of one acetyl group (B’-3OAc = 2.27 ppm). Integration of the anomeric protons demonstrated an equimolar occurrence of the three residues A, B and C in the alkali-treated polysaccharide. Analyses of the 1D and 2D NMR spectra revealed that the 〈-d-galactose (A) and the *N*-acetyl-^®^-d-glucosamine (C) were not modified by the alkaline treatment and that their respective ratio were also conserved. However, the amount of B residue found in the alkali-treated polysaccharide was equal to the sum of the B and B’ residues, which disappeared after alkaline treatment.

The chemical shifts of the ring protons of the B residue were determined using COSY (Appendix A) and TOCSY analyses (Table 1). The chemical shifts of the carbon of the residue were deduced by studying the *J^1^* and *J^3^* carbon–proton correlations using HMBC and HSQC spectra (Appendix A, Table 1). In the HMBC spectra (Figure 2A), the strong correlation between the proton B-H5 = 4.99 ppm with a weak signal attributed to B-C6 at 175.0 ppm showed that the B residue carried a carboxyl group. The chemical shift of the carbon B-C2 = 47.06 ppm and the correlations of the carbonyl at 174.47 ppm with the proton B-H2 = 4.45 ppm and the methyl of the acetyl group at 2.17 ppm (correlated with the signal at 22.59 ppm in the HSQC spectrum, Appendix A) showed that the B residue was a *N*-acetylated osamine (Figure 2A). The B-C2 chemical shift at 47.06 ppm was similar other published data of GulNAcA [10,11,12,13,14,15]. The C-2 NMR signal of the stereoisomers other than the *gulo* configuration showed a significantly lower-field position [16]. This hypothesis was also supported by comparison of calculated and observed chemical shifts. Calculations were conducted according to the method of Lipkind and co-workers [13]. Chemical shifts used for free l-GulNAc used as reference were found in [17]. Altogether, the values recorded and calculated (Appendix A) showed that the B residue had all the characteristics of a 2-acetamido-2-deoxy-〈-l-guluronic acid (〈-l-GulNAcA).

The alkaline treatment of the polysaccharide led to the removal of the acetyl group (3-OAc-H = 2.27 ppm, 3-OAc-C = 21.20 ppm), which is accompanied by an increase in the amount of the B residue. Therefore, we hypothesized the B’ residue was an acetylated form of the B residue. The proton and carbon chemical shifts of the B’ residue determined by mono- and two-dimentional NMR experiments (Table 1) revealed that they were very similar to that of the B residue, except for the carbon B’-C3 = 67.84 (B-C3 = 65.79 ppm) and the proton B’-H3 = 5.26 ppm (B-H3 = 4.18 ppm) which differed substantially. In fact, the carbonyl of the acetyl group B’-3OAc at 174.00 ppm correlated with the proton B’-H3 = 5.26 ppm and the methyl group at 2.27 ppm (Figure 2B), showing that the B’ residue was O-acetylated at the position 3. We concluded that the acetylation in the position 3 of the B residue gave the B’ residue: 3O-acetyl-2-acetamido-2-deoxy-〈-l-guluronic acid (3OAc-GulNAcA). Integration of the anomeric signals suggested a B:B’ ratio of 0.7:0.3 in the native polysaccharide. Finally, the native polysaccharide was made of four residues: 〈-d-galactose, 〈-l-GulNAcA/〈-l-3OAc-GulNAcA and ^®^-d-GlcNAc.

### 2.2. Structure of Oligosaccharides

The unmodified polysaccharide was subjected to mild acid hydrolysis with 0.1 M TFA. The resulting mixtures of oligosaccharides prepared were very complex and it was therefore very difficult to isolate pure oligosaccharides for in depth NMR analyses. We found that the mild acid degradation of the alkali treated polysaccharide―the deacetylated form―gave less complex mixtures which allowed the isolation of several pure oligosaccharides. The degradation products were fractionated, at first, by permeation gel chromatography (Figure 3A,B). At this step, in the case of a polysaccharide incubation of 90 min, one collected fraction contained a pure trisaccharide (Figure 3A). The other fractions contained oligosaccharides with the same degree of polymerization (e.g. di-, trisaccharides) but with various degrees of acetylation. Indeed, in order to obtain sufficiently large amounts of small oligosaccharides, the prolonged acid hydrolysis treatment required for the cleavage of the glycosidic bond has also lead to the removal of some *N*-linked acetyl groups. A second step of purification by semi-preparative anion exchange was necessary to isolate pure oligosaccharides. As illustrated in Figure 3, the fractions A and B collected from the degradation products of the 300 min hydrolysis (Figure 3B) contained several oligosaccharides that could be isolated (Figure 3C,D).

At the end, we collected two disaccharides and two trisaccharides in sufficient amounts for detailed analyses by NMR (^1^H spectra in Figure 4). Proton and carbon signals were ascribed straightforwardly combining COSY (Appendix A), HSQC (Appendix A) and HMBC experiments and values reported in Table 2. The data recorded for the first disaccharide (Figure 4A) were in agreement with a disaccharide made of GulNAcA (residue B) linked to GlcN (deacetylated residue C, C^da^) and allowed confirming the occurrence of 2-acetamido-2-deoxy-〈-l-guluronic acid (GulNAcA) in the polysaccharide. The HMBC spectrum recorded on this disaccharide (data not shown) was better resolved than that of the native polymer and clearly revealed the linkage between B and C^da^ residues. The correlations between the carbon B-C1 (99.10 ppm) with the protons C^da^ -H4^®^ (3.62 ppm) and C^da^ -C4〈/^®^ (78.86/78.95 ppm) with the proton B-H1 (5.08 ppm) demonstrated the GulNAcA residue was bound to the GlcN residue by a 〈(1,4) linkage. The structure of the second disaccharide (Figure 4B) was also thoroughly examined. In contrast with the previous oligosaccharide, we observed a deacetylation on the B residue giving 2-amino-2-deoxy-〈-l-guluronic acid (GulNA, B^da^) but not deacetylation on the C residue (GlcNAc). Again, the 〈(1,4) linkage was confirmed using HMBC experiments.

More interestingly, we purified a trisaccharide corresponding to the repetition unit of the polysaccharide eluting at 455 min in permeation gel chromatography (Figure 3A). As for the analyzed disaccharides, the C residue (GlcNAc) was located at the reducing end and the proton and carbon chemical shifts were very similar (Table 2). The ring proton of the B residue as well as the corresponding carbon were easily attributed after combining the COSY and HSQC experiments (Figure 5). The chemical shifts of the protons B-H3, B-H4 and B-H5 were higher than in the disaccharides, in agreement with its internal location in the trisaccharide instead of at the non-reducing end in the disaccharide. The 〈(1,4) linkage between the B and C residues was validated by HMBC (Figure 6C): the correlations between the carbon B-C1 (99.06 ppm) with the protons C-H4〈/^®^ (3.69/3.65 ppm) and C-C4〈/^®^ (79.15/78.97 ppm) with the proton B-H1 (5.08 ppm) were well-identified. The expected chemical shifts of the residue A (〈-d-Gal) were measured on the NMR spectra. The resolution of the HMBC spectra allowed clearly visualizing the linkage between the A and B residues. The anomeric carbon of the 〈-Gal A-C1 (96.84 ppm) correlated with the proton B-H4 (4.24 ppm) and the carbon B-C4 (74.82 ppm) correlated with the proton A-H1 (5.13 ppm), demonstrating the 〈(1,4) linkage between the two residues. Altogether, the data recorded were in agreement with a trisaccharide having the structure: 〈-d-Gal-(1→4)-〈-l-GulNAcA-(1→4)-〈/^®^-d-GlcNAc.

MALDI-TOF Mass spectrometry experiments conducted in positive ionisation mode with the fraction enriched in the trisaccharide (Appendix A) confirmed the expected molecular mass of the trisaccharide under the sodium form of the uronic acid function [M + Na]^+^ (*m*/*z* = 623). The occurrence of the molecular species [M – H + 2Na]^+^ is characteristic of an additional sodium ion interaction with the carboxylic function present in the oligosaccharide. Observation of such species confirmed the carboxylic function and excluded amide function, for example. A deacetylated form of the trisaccharide and two disaccharides were also observed in this fraction and their measured masses were also in agreement with the determined structure.

### 2.3. Structure of the Polysaccharide

Analyses of the native and alkali-treated polysaccharides, as well as the detailed characterization of purified di- and tri-saccharides, demonstrated that the polysaccharide is made of a trisaccharide repetition unit composed of 〈-d-galactose, 〈-l-GulNAcA (or 〈-l-3OAc-GulNAcA) and ^®^-d-GlcNAc. Determination of the linkages between the residues were facilitated by the preparation of the pure oligosaccharides. However, we were not able to isolate oligosaccharides having the C residue (d-GlcNAc) linked to the A residue (〈-d-Gal). Using the data accumulated by full analyses of the oligosaccharides, the NMR spectra of the polysaccharide and, more especially, the HBMC spectra were re-examined to highlight the connectivity between the C and A residues.

The correlations observed between the A and B residues, and B and C residues, in the oligosaccharides HMBC spectra were also observed in the HMBC spectra of the polysaccharides (Figure 6). The linkage between the C and A residues were also clearly identified: the A-C3 carbon (80.19 ppm) correlated with the C-H1 proton (4.81 ppm) (Figure 6) and reciprocally the C-C1 carbon of the native polysaccharide (103.19 ppm) correlated with the A-H3 proton (3.99 ppm) (not shown), demonstrating that the ^®^-d-GlcNAc (residue C) is linked to the 〈-d-Gal (residue A) via a 1,3 linkage. The structure of the polysaccharide was therefore: [→3)-〈-d-Gal-(1→4)-〈-l-GulNAcA/〈-l-3OAc-GulNAcA-(1→4)-^®^-d-GlcNAc-(1→]. This repetition unit was very similar to that found in the lipopolysaccharide extracted from *Pseudoalteromonas nigrifaciens* KMM161 [15]. The ^13^C chemical shifts reported were in agreement with our analyses and supported our conclusions

In conclusion, the structure investigated in this study is the third exopolysaccharide structure secreted by a *V. alginolyticus* strain [7,8]. The composition of the repetition unit determined as well as the sequence of residues have no similarities, suggesting that the biosynthetic pathways of these polysaccharides probably have no common ancestor. The structural diversity found in *V. alginolyticus* strains is also true for all the strains of *Vibrio* genus. The Carbohydrate Structure Database [1,2] has 114 entries describing polysaccharides—including secreted polysaccharides or lipopolysaccharides—of the genus *Vibrio*. The structural diversity of the polysaccharides suggests a very high plasticity of the polysaccharide biosynthesis pathway of the strains belonging to the genus *Vibrio*.

## 3. Materials and Methods 

### 3.1. Production, Isolation and Purification of the Vibrio alginolyticus Exopolysaccharide (VA-EPS)

VA-EPS was produced by *Vibrio alginolyticus* (CNCM I-5035) in a 30 L fermenter containing marine broth medium (30 g/L sea salts, 1 g/L yeast extracts, 4 g/L peptone) supplemented with glucose (30 g/L) at 25 °C. The culture medium was inoculated at 10% (*v*/*v*) with a bacterial suspension in the exponential growth phase. The pH was adjusted and maintained at 7.2 by automatic addition of 1 M NaOH. The medium was oxygenated at 15 L/min with an agitation rate of 350 rpm. After 72 h of fermentation, bacterial cells were removed from the culture medium by centrifugation (16,000× *g*, 30 min). The supernatant, containing the excreted VA-EPS, was then purified by filtration through a cellulose membrane (0.7 µm) and then by ultrafiltration (100 kDa). The filtration steps led to a loss of 20 to 30% biomass giving a purified 1 g/L VA-EPS in water. The sample was freeze-dried and stored at room temperature away from light and moisture.

### 3.2. Monosaccharide Analysis

The molar ratio of monosaccharides was determined according to [18], modified by [19]. The EPS was hydrolyzed with 3 M MeOH/ HCl at 110 °C for 4 h, followed by re-*N*-acetylation with Ac2O overnight at room temperature. The methyl glycosides were converted to their corresponding trimethylsilyl derivatives. Separation and quantification of the per-O-trimethylsilyl methyl glycosides were performed using gas–liquid chromatography (GLC) on an Agilent system equipped with a HP-5 ms capillary column (Agilent 0.25 mm × 30 m). The trimethylsilyl derivatives were analyzed using the following temperature program: 120 °C for 1 min, 120 °C→180 °C at 3 °C/min, 180 °C to 200 °C at 3 °C/min, 200 °C for 5 min.

### 3.3. Methylation Analysis

Glycosyl-linkage positions were determined as described in [20]. The native EPS was carboxyl-reduced by treatment with N-cyclohexyl 1-N’[^®^(N-methyl-morpholino)-ethyl] carbodiimide p-toluene sulfonate and with NaBD_4_ for 4 h at room temperature [21]. After dialysis against distilled water, hydroxyl groups were methylated using 2.5 N butyl lithium in hexanes and methyl iodide in DMSO [22]. The methylated compounds were extracted with CH_2_Cl_2_. The methylated products were then hydrolyzed in 2 M TFA for 2 h at 120 °C, then reduced with NaBD_4_ in a NH_4_OH solution for 30 min at 80 °C, and finally acetylated with 200 µL of 1-methyl imidazole and 2 mL of pyridine for 10 min at room temperature. GLC-mass spectrometry (MS) was performed on an Agilent instrument fitted with a high-performance 5 ms capillary column (Agilent, 0.25 mm × 30 m). The temperature program was 90 °C for 1 min, 90 °C to 300 °C at 5 °C/min, 300 °C for 1 min. Ionization was carried out in electron impact mode (EI, 70 eV).

### 3.4. Determination of Absolute Configuration

Assignment of absolute configuration of monosaccharide residues was adapted from the method of Gerwig [23,24]. A quantity of 2 mg of polysaccharide was dissoved in 500 µL of 4N TFA and maintained, in sealed glass tubes, for 4 h at 100 °C. After cooling, TFA was evaporated under a flux of nitrogen. Next, 500 µL of (S)-(+)-2-butanol and a drop of 13N TFA were added to the dried sample and the hermetically sealed glass tube was kept at 8 hours at 80 °C. Butanol and TFA were evaporated under nitrogen. Butylglycosides samples were then re-N-acetylated, converted to their corresponding trimethylsilyl derivatives and analyzed by GLC according to the protocol described in Section 3.2. for monosaccharide analysis.

### 3.5. Molecular Weight Determination

The molecular weight of VA-EPS was determined by high-performance size-exclusion chromatography (HPSEC) using an eighteen-angle light scattering detector, coupled with refractive index detection and specific refractive index increment dn/dc (DAWNTM HELEOS, Wyatt). Elution was performed on Shodex OHpak SB-805 HQ and OHpak SB-806 HQ placed in series (Phenomenex, exclusion limit <2 × 10^7^ g/mol) with 0.1 M NaNO_3_ as the eluent. To calculate the molecular mass, the dn/dc value used was 0.145 mL/g. The polydispersity index was calculated from the Mw/Mn ratio.

### 3.6. Acid Hydrolysis and Oligosaccharides Purification

VA-EPS underwent mild acid hydrolysis. A quantity of 300 mg of polysaccharide was solubilized in 75 mL of 0.1 M TFA and heated at 100 °C for 90 min or 300 min, respectively. After neutralization with 8N NH_4_OH, the salts were eliminated by adding five volumes of acetone. The precipitate that contained the oligosaccharides was recovered by centrifugation. The pellet was resuspended in 2 mL of distilled water and the oligosaccharides were fractionated by size exclusion chromatography using a SEC Toyopearl HW-40 column (5 × 100 cm, Tosoh, exclusion limit <10^4^ Da) with 0.1 M (NH_4_)_2_CO_3_ as the eluent.

### 3.7. NMR

Carbon-13 and proton NMR spectra were recorded with a Bruker Avance 400 spectrometer operating at a frequency of 100.618 MHz for ^13^C and 400.13 MHz for ^1^H. Samples were solubilized in D_2_O at a temperature of 293 K for the oligosaccharides and 353 K for the polysaccharide. Residual signal of the solvent was used as internal standard: HOD at 4.85 ppm at 293 K and 4.35 ppm at 343 K. ^13^C spectra were recorded using 90° pulses, 20,000 Hz spectral width, 65,536 data points, 1.638 s acquisition time, 1 s relaxation delay and between 8192 and 16,834 scans. Proton spectra were recorded with a 4006 Hz spectral width, 32,768 data points, 4.089 s acquisition time, 0.1 s relaxation delay and 16 scans. The ^1^H and ^13^C-NMR assignments were based on ^1^H-^1^H homonuclear and ^1^H-^13^C heteronuclear correlation experiments (correlation spectroscopy, COSY; heteronuclear multiple-bond correlation, HMBC; heteronuclear single quantum correlation, HSQC). They were performed with a 4006 Hz spectral width, 2048 data points, 0.255 s acquisition time, 1 s relaxation delay; 32 to 512 scans were accumulated.

## Figures and Tables

**Figure 1 marinedrugs-18-00509-f001:**
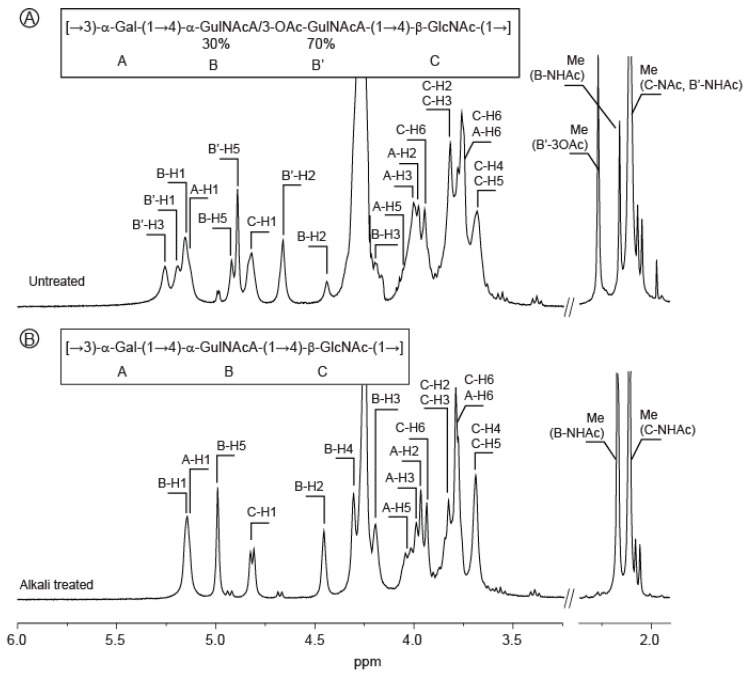
^1^H NMR spectra of the *Vibrio alginolyticus* exopolysaccharide. The spectra of the purified polysaccharides were recorded at 353 K prior to (**A**) or after alkaline treatment (**B**). Inset: chemical structure of the polysaccharide.

**Figure 2 marinedrugs-18-00509-f002:**
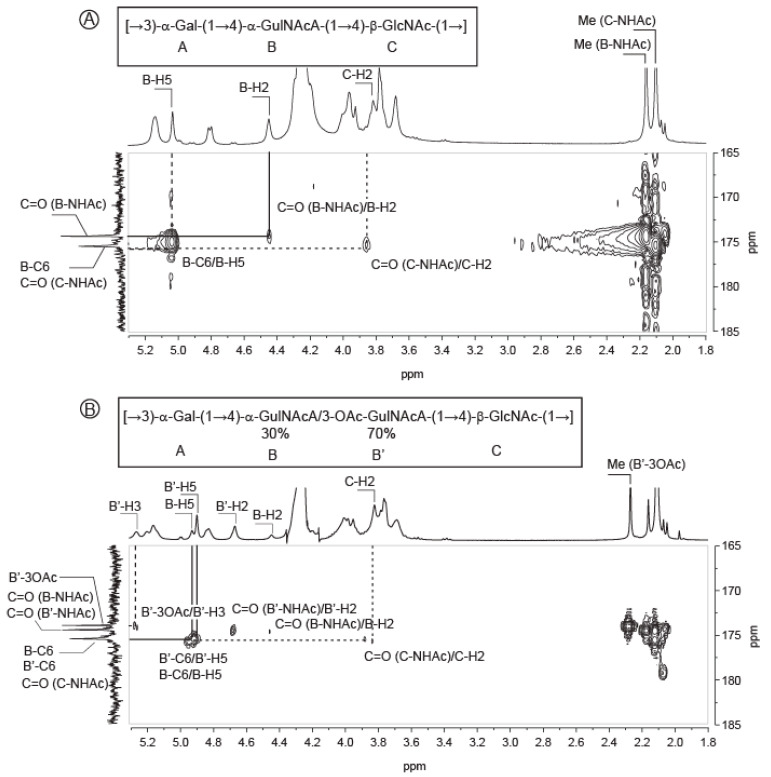
Detail of the HMBC spectra recorded at 353K on *Vibrio alginolyticus* exopolysaccharide highlighting the acetylation of the guluronic acid residue. (**A**) HMBC of the deacetylated exopolysaccharide at the position 3 of the GulNAcA; (**B**) HMBC of the untreated exopolysaccharide.

**Figure 3 marinedrugs-18-00509-f003:**
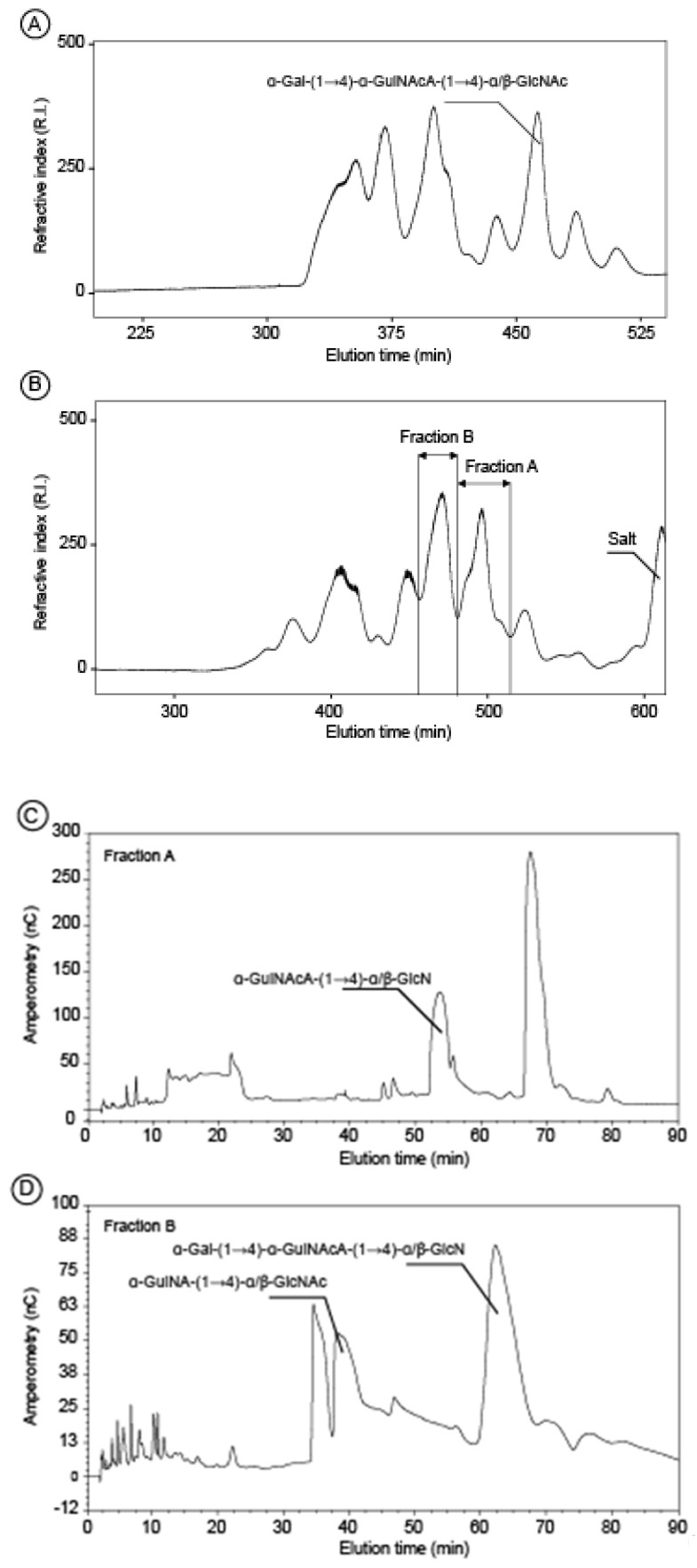
Chromatography experiments leading to the preparation of pure oligosaccharides (**A**) and (**B**) gel permeation chromatograms recorded on a mixture of oligosaccharides prepared by mild acid hydrolysis of *Vibrio alginolyticus* exopolysaccharide in 0.1M TFA at 100 °C for 90 min and 300 min, respectively. (**C**) and (**D**) Fractions A and B of the chromatogram (**B**) were further fractionated by anion exchange chromatography allowing purification of two disaccharides and one trisaccharide.

**Figure 4 marinedrugs-18-00509-f004:**
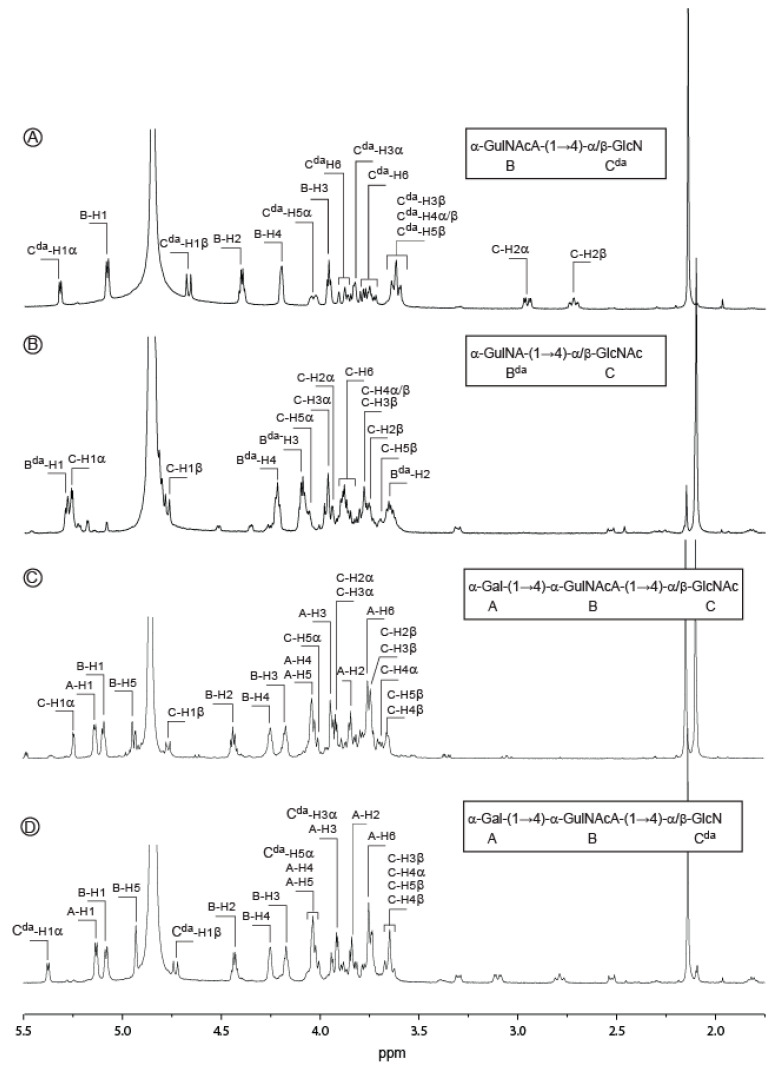
^1^H NMR spectra recorded at 293K of the oligosaccharides purified by chromatography. The annotated spectra correspond to two disaccharides (**A**,**B**) and two trisaccharides (**C**,**D**). Inset: chemical structure of the oligosaccharides.

**Figure 5 marinedrugs-18-00509-f005:**
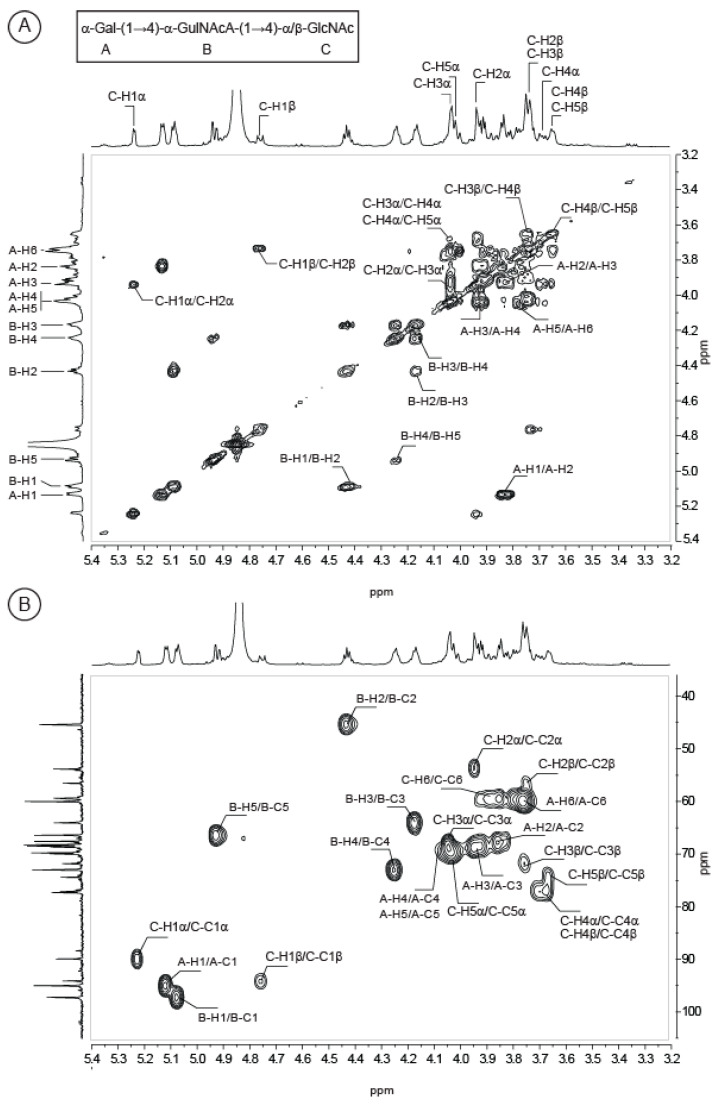
COSY (**A**) and HSQC (**B**) spectra recorded at 293K of the purified trisaccharide, whose structure corresponds to the repetition unit of the *Vibrio alginolyticus* exopolysaccharide.

**Figure 6 marinedrugs-18-00509-f006:**
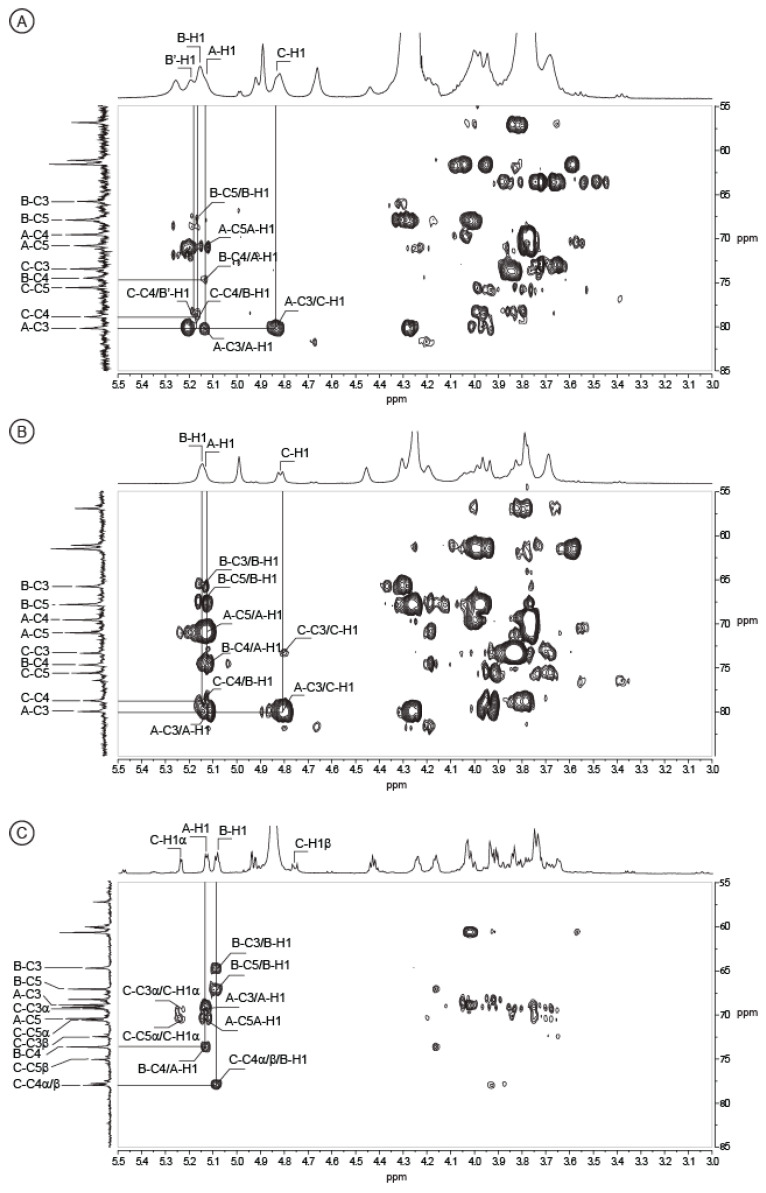
Detail of HMBC spectra. The spectra were recorded at 353K on the complete polysaccharide (**A**) and after alkali-treatment (**B**), and at 293K on the trisaccharide (**C**). ^1^H/^13^C heteronuclear correlations that helped to determine linkages between residues are indicated in the spectra.

**Table 1 marinedrugs-18-00509-t001:** ^1^H and ^13^C NMR chemical shifts (δ, ppm) of the *Vibrio alginolyticus* exopolysaccharide before and after alkaline treatment.

Sugar Residue		1	2	3	4	5	6(6a,6b)	NHAc (CH3,CO)	OAc (CH3,CO)
EPS									
→3)-〈-d-Gal-(1→	^1^H	5.13	3.95	3.99	4.26	4.05	3.77,3.93		
	^13^C	96.66	67.62	80.19	69.58	70.82	61.58		
→4)-〈-l-GulNAcA-(1→	^1^H	5.14	4.44	4.18	4.30	4.92		2.16	
	^13^C	98.83	46.99	65.79	74.49	67.92	175.40	22.59, 174.47	
→4)-〈-3OAc-l-GulNAcA-(1→	^1^H	5.18	4.66	5.26	4.31	4.89		2.11	2.27
	^13^C	97.1	45.62	67.84	71.68	68.1	175.75	22.59, 174.47	21.20, 174.00
→4)-^®^-d-GlcNAc-(1→	^1^H	4.81	3.81	3.79	3.68	3.68	3.77,3.94	2.11	
	^13^C	103.19	56.84	73.45	78.89	75.58	61.13	22.99, 175.50	
Deacetylated EPS									
→3)-〈-d-Gal-(1→	^1^H	5.14	3.95	3.95	4.25	4.00	3.77,3.93		
	^13^C	97.01	67.81	79.95	69.56	71.02	61.49		
→4)-〈-l-GulNAcA-(1→	^1^H	5.14	4.45	4.19	4.30	4.99		2.17	
	^13^C	99.04	47.06	65.76	74.66	67.81	175.00	22.59, 174.47	
→4)-^®^-d-GlcNAc-(1→	^1^H	4.81	3.81	3.81	3.68	3.68	3.77,3.93	2.11	
	^13^C	103.1	56.92	73.33	78.71	75.59	61.08	22.99, 175.59	

**Table 2 marinedrugs-18-00509-t002:** ^1^H and ^13^C NMR chemical shifts (δ, ppm) of the *Vibrio alginolyticus* oligosaccharides purified after alkaline treatment and mild acid hydrolysis.

Sugar Residue		1	2	3	4	5	6(6a, 6b)	NHAc (CH_3_,CO)
Trisaccharide ABC								
〈-d-Gal-(1→	^1^H	5.13	3.84	3.93	4.04	4.02	3.74, 3.74	
	^13^C	96.84	69.42	70.31	70.07	71.61	61.83	
→4)-〈-l-GulNAcA-(1→	^1^H	5.08	4.43	4.17	4.24	4.93		2.14
	^13^C	99.06	47.28	65.89	74.82	68.25	175.89	22.93, 175.13
→4)-〈-d-GlcNAc	^1^H	5.24	3.93	4.03	3.69	4.02	3.76, 3.85	2.09
	^13^C	91.76	55.66	70.48	79.15	71.79	61.20	23.05, 175.66
→4)-^®^-d-GlcNAc	^1^H	4.76	3.73	3.74	3.65	3.64	3.76, 3.85	2.09
	^13^C	95.97	58.37	73.65	78.97	76.24	61.33	23.33, 175.89
Trisaccharide ABC^da^								
〈-d-Gal-(1→	^1^H	5.13	3.84	3.93	4.04	4.03	3.75, 375	
	^13^C	96.78	69.36	70.24	69.98	71.55	61.75	
→4)-〈-l-GulNAcA-(1→	^1^H	5.08	4.43	4.17	4.25	4.93		2.14
	^13^C	99.03	47.19	65.78	74.74	68.19	176.88	22.86, 175.08
→4)-〈-d-GlcN	^1^H	5.37	3.10	3.91	3.65	4.04	3.76, 3.87	
	^13^C	91.73	56.18	71.36	78.69	71.91	61.24	
→4)-^®^-d-GlcN	^1^H	4.73	2.79	3.66	3.65	3.65	3.76, 3.87	
	^13^C	96.46	58.76	74.47	78.69	76.29	61.05	
Disaccharide B^da^C								
〈-l-GulNA-(1→	^1^H	5.27	3.64	4.10	4.21	4.82		
	^13^C	97.85	48.17	69.56	70.97	68.93	177.22	
→4)-〈-d-GlcNAc	^1^H	5.25	3.96	3.97	3.77	4.06	3.87, 3.92	2.09
	^13^C	91.63	55.41	70.31	79.28	71.33	61.80	23.05, 175.70
→4)-^®^-d-GlcNAc	^1^H	4.77	3.75	3.77	3.77	3.68	3.87, 3.92	2.09
	^13^C	95.92	58.11	73.54	79.15	75.83	61.91	23.33, 175.96
Disaccharide BC^da^								
〈-l-GulNAcA-(1→	^1^H	5.08	4.40	3.96	4.20	4.85		2.14
	^13^C	99.10	47.10	70.34	71.29	68.75	177.48	22.84, 175.05
→4)-〈-d-GlcN	^1^H	5.31	2.95	3.82	3.62	4.03	3.74, 3.87	
	^13^C	92.47	56.29	72.41	78.86	71.73	61.25	
→4)-β-d-GlcN	^1^H	4.66	2.72	3.62	3.62	3.64	3.74, 3.87	
	^13^C	97.09	58.81	74.93	78.95	76.19	61.05	

da: deacetylated.

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
