# Peer review of "Structure of the Polysaccharide Secreted by Vibrio alginolyticus CNCM I-5035 (Epidermist 4.0TM)"

_marinedrugs, 2020, doi:10.3390/md18100509_

Round 1
Reviewer 1 Report
Accept
Reviewer 2 Report
The authors have added new experimental information and responded adequately the raised questions. The overall manuscript has been improved with the corrections and modifications now introduced.
This manuscript is a resubmission of an earlier submission. The following is a list of the peer review reports and author responses from that submission.
Round 1
Reviewer 1 Report
The work by Drouillard et al. is esssentially an analytical work. It presents the structural characterization by NMR of an exopolisaccharide produced by a marine bacteria.
As it is now, it only shows NMR characterization data but it lacks of the methylation analysis and absolute configuration data although are mentioned in the experimental part.
From the data of neutral monosaccharide analysis and the NMR point of view, the Galactose and GlcNAc residues are confidently characterized. The less common residue, N-Acetylaminoguluronic acid and its O-acetylated variant are only supported by one singular chemical shift (C2 bound to N).
To support this conclusion it is recomended to bring more inforamtion or data. More information in the introduction regarding why it is expected to find guluronic acid derivative in this EPS will be recomendable.
is it possible to compare with an availble standard of this type of monosaccharide?, Can be obtained and presented MS data derived from monosaccharide compositional analysis that suppors the presence og GulNAcA or derivatives?.
Additionally, CSDB is cited in the first 3 references. However, have try to use the authors the CSDB on-line tools for NMR prediction from structure information (and viceversa, structure proposal from NMR data) to support their conclusion?.
Some minor technical points.
Some times is said 0.1 N TFA and others 0.1 M TFA, better use one expression.
Figure 3 A and B, better to present them with the same X-scale for easy comparison.
Figure 4 A: Is it taken at the same temperature than others. It seems that the signals are shifted. Signals for H2-GlcN anomers seem to be too much shifted to high field . Probably their chemical shift will be also variable depending of the pH of the sample.
Figure 6: the figure only shows the H1-C3 correlation but not the H3-C1 , the 13C spectral window does not include C1. Some signals mentioned in the text are not presented in the spectrum.
Finally a question regarding possible "conflict of interest":
The studied EPS is a component of "epidermist 4.0 TM", 2 lines of 8 in the abstract are dedicated to applications of that propietary product but the work is not related at all with its properties and applications. Some of coauthors are from "Polymaris Biotechnology"" and "CODIF International". Are these companies propietaries or vendors of that product?. If so, best to indicate in "conflict of interest". When "epidermist 4.0 TM" is mention in the manuscript, it is referenced to "Encyclopedia of polymer applications" it will be useful to cite also any patent registry or propietary document that could exist. Additionally in the epxerimental part, in the producction of the EPS, information regarding the fermenter should be provided.
Reviewer 2 Report
The authors isolated, purified and characterized an exopolysaccharade excreted by Vibrio alginolyticus. They managed to identify not only the composition of the repeating unit but also a linkage among the units. They worked with raw polysaccharides, the polysaccharides that were hydrolyzed by acid hydrolysis, and artificially prepared oligosaccharides. The principal instrumental analysis was 1H and 13C NMR spectroscopy.
According to my view it is a thoroughly done structural analysis with a brave combination of methods. If I were to do such an analysis, I would combine the results of NMR experiments with HR MS spectrometry or LC-MS, which might extend the portfolio of experimental results.
The manuscript is well written with a few typographical errors or incorrect expressions. The list of them follows:
- 22: tree – three
- 41: spectroscopy methods – spectroscopic methods
l.41: methods stay… - methods follows…
- 56: cosmetics sectors – cosmetic sectors
- 56: …the elucidation of two EPSs.. - ….the elucidation of structures of two EPSs..
- 57: cosmeceutical – cosmetic
- 60-62: The sentence should be reformulated, it does not make any sense to me.
- 66: we undertaken- we have undertaken
- 119: bi-dimensional NMR – two dimensional NMR
- 155: enough large amount – sufficiently large amount
- 161: enough amount – sufficient amount
- 171: a second – the second
- 172: fully examined – thoroughly examined
Reviewer 3 Report
The paper of Sophie Drouillard et al. describes the structure elucidation of an EPS structure isolated from Vibrio alginolyticus CNCM I-5035.The structure has been determined using proper techniques.
I have a few comments to be addressed in the revised version of the manuscript:
- Manuscript Presentation
1) The conclusion part is absent. In my opinion, the authors should provide some examples of other (similar or produced by Vibrio sp.) polysaccharides described elsewhere.
2) In my opinion, authors should add a sentence or paragraph about the feasibility of obtaining oligosaccharides (what is the purpose?). Why was the sequence not initially established on the polysaccharide? Why was the ROESY experiment not used?
- Structure elucidation
1) L-GulNAcA is often found in the amide form. The values of H-5 chemical shifts of L-GulNAcA may correspond to the amide form. For example [Carbohydrate Research 338 (2003) 1251–1257, Carbohydrate Research 349 (2012) 78–81]. I recommend that authors check this out.
2) Line 114-115. It is necessary to add a more detailed description of how the absolute configuration was established. What oligosaccharide fragments and effects have been considered.
3) Line 237. Methylation analysis is not reported and not commented.
- Minor comments.
Line 91, 114, 127. The font size of the absolute configurations.
Line 115. Typo. Configuration.